# Age-Related Changes in Clinical and Analytical Variables in Chronic Hemodialyzed Patients

**DOI:** 10.3390/ijms25063325

**Published:** 2024-03-15

**Authors:** Luís Belo, Maria João Valente, Susana Rocha, Susana Coimbra, Cristina Catarino, Irina Lousa, Elsa Bronze-da-Rocha, Petronila Rocha-Pereira, Maria do Sameiro-Faria, José Gerardo Oliveira, José Madureira, João Carlos Fernandes, Vasco Miranda, José Pedro L. Nunes, Alice Santos-Silva

**Affiliations:** 1UCIBIO—Applied Molecular Biosciences Unit, Laboratory of Biochemistry, Department of Biological Sciences, Faculty of Pharmacy, University of Porto, 4050-313 Porto, Portugalassilva@ff.up.pt (A.S.-S.); 2Associate Laboratory i4HB—Institute for Health and Bioeconomy, Faculty of Pharmacy, University of Porto, 4050-313 Porto, Portugal; 3National Food Institute, Technical University of Denmark, 2800 Kongens Lyngby, Denmark; 41H-TOXRUN—One Health Toxicology Research Unit, University Institute of Health Sciences, CESPU, CRL, 4585-116 Gandra, Portugal; 5Health Sciences Research Centre, University of Beira Interior, 6200-506 Covilhã, Portugal; 6Hemodialysis Clinic of Felgueiras (CHF), 4610-106 Felgueiras, Portugal; 7Hemodialysis Clinic of Porto (CHP), 4200-277 Porto, Portugal; 8Center for Health Technology and Services Research (CINTESIS), Faculty of Medicine, University of Porto, 4200-319 Porto, Portugal; 9NefroServe Hemodialysis Clinic of Barcelos, 4750-110 Barcelos, Portugal; 10NefroServe Hemodialysis Clinic of Viana do Castelo, 4901-858 Viana do Castelo, Portugal; 11Hemodialysis Clinic of Gondomar, CHD, 4420-086 Gondomar, Portugal; 12Faculty of Medicine, University of Porto, 4200-319 Porto, Portugal

**Keywords:** end-stage renal disease, dialysis, aging, biomarkers, biological variables

## Abstract

Worldwide, the number of elderly individuals receiving chronic hemodialysis is rising. The aim of our study was to evaluate several clinical and analytical biomarkers in chronically dialyzed patients and analyze how they change with age. A cross-sectional study was performed by evaluating 289 end-stage renal disease patients undergoing dialysis. We evaluated the hemogram, adipokines, the lipid profile, and several markers related to inflammation, endothelial function/fibrinolysis, nutrition, iron metabolism, and cardiac and renal fibrosis. Clinical data and dialysis efficacy parameters were obtained from all patients. The relationships between studied biomarkers and age were assessed by a statistical comparison between younger (adults with age < 65 years) and older (age ≥ 65 years) patients and by performing regression analysis. Participants presented a mean age of 68.7 years (±13.6), with 66.8% (*n* = 193) being classified as older. Compared to younger patients, older patients presented the following: (a) significantly lower values of diastolic blood pressure (DBP) and ultrafiltration volume; (b) lower levels of phosphorus, uric acid, creatinine, and albumin; and (c) higher circulating concentrations of tissue-type plasminogen activator (tPA), D-dimer, interleukin-6, leptin, *N*-terminal pro B-type natriuretic peptide, and tissue inhibitor of metalloproteinase-1. In the multiple linear regression analysis, DBP values, tPA, phosphorus, and D-dimer levels were independently associated with the age of patients (standardized betas: −0.407, 0.272, −0.230, and 0.197, respectively; *p* < 0.001 for all), demonstrating relevant changes in biomarkers with increasing age at cardiovascular and nutritional levels. These findings seem to result from crosstalk mechanisms between aging and chronic kidney disease.

## 1. Introduction

Chronic kidney disease (CKD) is a major public health problem [1], and the incidence of elderly patients undergoing renal replacement therapy (RRT) is increasing worldwide [2]. Portugal has one of the highest incidences of patients undergoing RRT, with a high prevalence of older dialysis populations [3]. In this country, the prevalence of patients treated by hemodialysis has approximately doubled in the last twenty years, and the average age of the patients has increased continuously. In 2018, the mean age of patients treated by hemodialysis in Portugal was 68 years, and 62.8% were 65 years old or older [3].

Advanced age is a widely recognized risk factor for CKD. Conversely, CKD is a promoter of biological aging and is associated with reduced lifespan [4,5]. Many factors are involved in premature aging in CKD, including hormonal imbalance and glycative and oxidative stresses [4]. Actually, oxidative stress is one of the most accepted theories of human aging, though the different originally independent theories are being brought together into a single unified theory of aging [6]. A major goal of studying mechanisms of aging is to promote interventions that may increase human longevity.

In humans, chronological aging is a predictor of biological aging, but its interrelation is not linear [7]. Chronological age may differ significantly from biological age, particularly in the presence of some diseases, such as CKD. Likewise, age-related changes in biochemical variables are potentially influenced by disease states. The study of variables that are influenced by age may allow a better interpretation of the results in older patients, setting new reference values in some populations. For instance, the use of different hemoglobin cutoff levels for the (re)definition of anemia in elderly populations is highly debated [8]. The same applies to D-dimer values in diagnosing deep vein thrombosis in older patients [9].

In the present study, we evaluated a wide range of clinical and laboratory parameters to study their associations with aging in end-stage renal disease (ESRD) patients under chronic dialysis therapy. We included in the study several biomarkers frequently addressed in aging research, namely hemodynamic, hematological, inflammatory, nutritional, cardiac, and lipid variables.

## 2. Results

### 2.1. Demographic and Clinical Data of Patients

We studied 289 ESRD patients under chronic dialysis therapy with a mean age of 68.7 ± 13.6 years (range 23–94); 193 patients (66.8%) were classified as older and 96 (33.2%) as younger. Younger and older patients presented similar gender distribution and body mass index (BMI) values (Table 1).

Patients were under dialysis therapy three times per week, for a median period of 4 h, and used high-flux polysulfone FX-class dialyzers (1.4–2.2 m^2^) of Fresenius (Bad Homburg, Germany). The most prevalent dialysis modality used by patients was online hemodiafiltration, and arteriovenous fistula was the most prevalent vascular access used. The distribution of dialysis therapy and vascular access modalities did not differ between younger and older groups.

The main causes of renal failure were diabetes mellitus (*n* = 101) and uncertain etiology (*n* = 66), followed by other diseases (*n* = 44), arterial hypertension (*n* = 36), glomerulonephritis (*n* = 23), and polycystic renal disease (*n* = 19). Regarding CKD etiology, older patients presented a higher percentage of diabetic nephropathy cases and lower chronic glomerulonephritis cases.

A total of 179 patients (61.9%) had arterial hypertension and 124 (42.9%) were diabetic. The number of hypertension cases did not differ between groups, but the elderly group presented a significantly higher number of diabetic cases (47.7% vs. 33.3%, *p* = 0.023).

The most frequent pharmacological therapies were as follows: ESA (*n* = 240, 83%), intravenous iron (*n* = 186, 64.4%), phosphate binders (*n* = 165, 57.1%), statins (*n* = 157, 54.3%), antihypertensives (*n* = 126, 43.6%), and antidiabetics (*n* = 112, 38.8%). The relative number of patients under these pharmacological therapies was similar between both groups, except for antihypertensive therapy, which was higher for younger patients (53.1% vs. 38.9%, *p* = 0.024). ESA and iron doses were also similar between the two groups (Table 1).

Older patients presented lower diastolic blood pressure (DBP) and ultrafiltration volume values, lower phosphorus, uric acid, creatinine, albumin, and urea, and higher serum calcium concentrations (Table 1).

### 2.2. Hematological and Biochemical Data of Patients

Compared to younger patients, older dialysis patients presented higher circulating concentrations of tPA, D-dimer, IL-6, leptin, NT-proBNP, and TIMP-1 (Table 2). The TNF-α concentration was also higher in older patients but did not reach statistical significance (*p* = 0.066).

The age of the patients correlated positively and significantly with tPA, D-dimer, NT-proBNP, TIMP-1, and IL-6 levels and was inversely and significantly correlated with DBP, ultrafiltration volume, and levels of phosphorus, creatinine, uric acid, albumin, urea, and triglycerides (Table 3). These correlations remained statistically significant when analyzed separately by gender, except for TIMP-1 in men (borderline statistically significant) and triglycerides in women. The association of age with DBP was independent of antihypertensive therapy (Figure 1A) and with phosphorus levels was independent of phosphate binder use (Figure 1B). In the multiple linear regression analysis, DBP and circulating levels of tPA, phosphorus, and D-dimer remained statistically associated with age (Table 4).

Within all patients, phosphorus levels also correlated positively and significantly with serum uric acid (*r* = 0.330, *p* < 0.001) and with creatinine (*r* = 0.431, *p* < 0.001) levels. t-PA levels were positively correlated with D-dimer (*r* = 0.223, *p* < 0.001); BMI was positively correlated with leptin levels (*r* = 0.544, *p* < 0.001) and negatively with adiponectin (*r* = −0.402, *p* < 0.001).

## 3. Discussion

CKD is a promotor of biological aging due to several mechanisms [4], and CKD patients present an increased risk of mortality, particularly due to cardiovascular disease [10]. The identification of age-related changes in biomarkers may provide a better clinical evaluation of old patients. Our results show that in ESRD patients under chronic dialysis, DBP and levels of tPA, phosphorus, and D-dimer are independently associated with age.

We studied patients from different clinics in Portugal. Our study reveals a high percentage of elderly patients under chronic hemodialysis (66.8%) according to our cutoff (65 years), which was the same used by Tufan et al. [11]. Furthermore, the mean age of overall participants (68.7 years) is comparable to the median age (66 years) reported by Snaedal et al. for patients recruited from dialysis units in Sweden [12]. Similarly to other studies, the main CKD etiologies were diabetes and hypertensive renal disease [10,12]. The increased number of comorbidities—namely dyslipidemia, diabetes, and arterial hypertension—is also in line with previous reports [10,12].

Most biomarkers that we found related to age in dialysis patients are also considered cardiovascular risk markers, and this may justify particularly enhanced cardiovascular morbimortality in advanced age patients. Profound changes occur in the cardiovascular system with aging, with characteristic changes in blood pressure—probably the most frequently measured cardiovascular biomarker [13]. Systolic blood pressure (SBP) rises with age, mainly due to increased arterial stiffness, whereas DBP increases until the fifth decade and slowly decreases after the age of sixty [14]. Actually, cardiac diastolic function has been shown to deteriorate with advancing age. In the present study, we found lower values of DBP in older patients and a strong negative correlation with age, irrespective of antihypertensive therapy, which may be associated with decreased elasticity of the aorta and of the arterial tree in general (Figure 2). This result, together with higher NT-proBNP levels in older patients, as well as a positive correlation of this cardiac biomarker with age, suggests a negative impact on the heart in aging dialysis patients. NT-proBNP can reliably detect the presence of cardiac diastolic dysfunction [15], and in a community-dwelling population, NT-proBNP was reported to increase possibly in response to age-related alterations, including diastolic and renal functions [16]. The association of aging with NT-proBNP and, particularly, with DBP that we found in this dialysis cohort might be highly relevant, as both are important risk factors for cardiovascular morbimortality. Among the Framingham Heart Study participants, DBP was inversely related to coronary heart disease risk in patients of older ages [17].

While it is widely acknowledged that SBP increases with age, our patient group did not exhibit a discernible increase in SBP with age. This lack of association is explained by the fact that most patients, in both groups, presented isolated systolic hypertension (more related to the disease than to age-related effects), and others had blood pressure control with proper dialysis (comprehending fluid and electrolytic control) and with antihypertensive therapy. Interestingly, older patients were less medicated with antihypertensives, though this may be partially explained by current guidelines suggesting a less aggressive treatment of hypertension for CKD patients [18]. In addition, older patients seem to better control fluid intake and/or have less appetite, as they presented lower ultrafiltration volume at dialysis compared with younger patients.

Aging is associated with modifications in the hemostatic system, with increased production of coagulation factors and decreased fibrinolytic activity [19,20]. Despite this, the tPA antigen can be used as a marker of endothelial (dys)function, and its levels were reported to correlate positively with increasing age [21,22], in agreement with the results of the present study. Our group has also previously described this association in a smaller group of ESRD patients [23]. It is important to emphasize that tPA levels were positively correlated with D-dimer levels, but this correlation was not strong (*r* = 0.223). Thus, tPA release from endothelial cells is unlikely to be explained solely to promote fibrinolysis but is possibly due to age-related endothelium dysfunction (Figure 2). The inclusion of both tPA and D-dimer as independent variables in the final regression model (association with age) agrees with this hypothesis.

D-dimer levels were increased in our studied patients and positively correlated with age. It is known that D-dimer levels increase with age, making its interpretation difficult when assessing thrombotic disorders in elderly patients [24]. D-dimer levels are a result of fibrin formation and turnover. In dialysis patients, raised D-dimer levels are a result of various stimuli, namely regular venous puncture and hemostasis, contact with foreign surfaces, and increased crosstalk with inflammation, among others. The association of D-dimer levels with the age of chronic dialysis patients may be due to aged-related aspects in the hemostatic system (including procoagulation status), justifying its inclusion in the regression model (Figure 2).

We found that advanced age is significantly associated with a reduction in phosphorus levels, regardless of phosphate binder therapy. This is important, as hypophosphatemia is associated with increased mortality in elderly dialysis patients, while hyperphosphatemia is associated with higher mortality risks transversely in all age groups [25]. Our results are in accordance with a previous study in HD patients [25], but the underlying mechanism of this association remains unclear. In normal physiology, the kidney is the primary site of regulation for phosphate homeostasis [26], and aging is associated with a decrease in the intrinsic capacity of the kidney to reabsorb phosphate [27]. However, in ESRD patients, this mechanism is unlikely to explain the main differences with aging, as all patients present with kidney failure. Our results can rather be explained by decreased dietary protein intake [28] and/or decreased intestinal phosphorus absorption with aging. Despite mechanisms of intestinal phosphorus absorption having not been extensively studied, it was shown that the percentage of intestinal phosphorus absorption efficiency decreases with age [29]. Moreover, in our study, phosphorus levels correlated positively with serum uric acid and with creatinine, which can be used as nutrition markers in these patients [30,31]. It should be emphasized that uric acid is associated with protein intake and that lower creatinine levels in patients undergoing hemodialysis are associated with lower muscle mass and malnutrition [11,32]. In spite of these interrelations (between phosphorus, uric acid, and creatinine), only phosphorus level was independently associated with the age of ESRD patients in the regression model.

Serum leptin levels have been proposed for use in nutritional assessment of elderly patients [33]. Leptin is an adipokine, and its levels have been reported to correlate with body mass index (BMI) and skinfold thickness [33]. In our study, despite leptin concentration being higher in older patients and showing a strong positive correlation with BMI, neither of these parameters were correlated with the age of patients. The increase in visceral adiposity with advanced age (not necessarily accompanied by an increase in BMI) may underlie the higher leptin values in older patients. Also, high IL-6 levels may increase the expression and release of leptin in adipose tissue [34].

It is well described that ESRD patients present with an enhanced inflammatory state that is a predictor of all-cause mortality [35]. Furthermore, the “normal” reference range of inflammatory variables in these patients is a matter of debate [12,36]. Data presented in this study are in line with raised inflammation in the dialysis population, as demonstrated by most of the studied inflammatory variables. IL-6 and hsCRP were particularly elevated, being three to four times higher than those reported for healthy populations [35]. Furthermore, we were able to demonstrate that the IL-6 level is increased in the older group and positively correlated with the age of the participants, which is in line with the “inflammaging” theory [37]. TNF-α levels were also higher in older patients but did not reach statistical significance. The high inflammatory status of dialysis patients that aggravates with aging underlies the disruption of health-related status, being particularly detrimental late in life. The lack of association of PTX3 with aging may be due to the pleiotropic nature of this inflammatory marker, i.e., its levels are determined by a wide range of factors, mitigating the effect of aging.

Statin therapy was quite frequent in our study (both groups), potentially justifying the lack of differences in lipid profile values, in particular cholesterol levels. Due to very high cardiovascular risk, CKD patients are often monitored for LDL-C levels, with strict target values [38]. Only triglycerides showed an inverse correlation with age, though it was relatively weak, only observed in men, and probably also related to malnutrition in older patients.

We were unable to find an association between age and hemoglobin levels, despite an inverse association having been described in healthy elderly men and women [39]. The hemoglobin values observed in our patients are conditioned by ESA and/or iron therapies, in order to correct anemia presented by these patients. Actually, the major cause of anemia in these patients is a lack of erythropoietin synthesis by failing kidneys. Furthermore, despite frequent disturbances in iron metabolism, mainly due to inflammation and hepcidin involvement [40], patients were adequately treated with IV iron. This is also in line with the lack of association between the age of participants and markers of iron metabolism. Older patients presented a lower sTfR value, though without statistical significance (*p* = 0.105), possibly revealing a decrease in erythropoiesis in these patients. The sTfR level is influenced by factors such as erythropoiesis and iron status [41,42], with both being conditioned by ESA and iron therapies, which were similar for both groups (Table 2). It is therefore not surprising that no differences were found for reticulocyte or RBC counts.

Hepcidin plays a central role in regulating iron availability, and its levels are increased in CKD patients due to the reduction in renal clearance [43], though a contribution of inflammation to its synthesis is likely to occur [40]. Despite raised inflammation in ESRD patients and a positive correlation of age with IL-6 levels, in the present study hepcidin levels presented no correlation with age. Hepcidin is possibly more influenced by iron status (and therapy) than age-related inflammation.

Finally, the TIMP-1 concentration was higher in older patients and positively correlated with age. This might result mostly from kidney disease itself and its duration, but a contribution of aging to renal fibrosis cannot be excluded [44]. Aging is associated with profound renal changes, including glomerulosclerosis and interstitial fibrosis [45]. The inhibition of metalloproteinases by TIMP-1 contributes to extracellular matrix deposition, leading to fibrosis [44]. In our study, as all patients presented with kidney failure, the magnitude of the association of TIMP-1 with aging was not very strong, and the variable showed no independent association with age in the multiple regression model.

Our study presented some limitations. Daily variations due to sample collections at different time periods (collected before dialysis sessions) could interfere with the values of some variables due to circadian variations. Samples were also collected on a non-fasting basis, affecting the analysis of a few parameters, namely triglycerides. Moreover, the studied patients presented several comorbidities and were strongly medicated, limiting the analysis of the aging-related effects on the studied variables. Some of the variables may be disease related.

In conclusion, we identified variables that are strongly influenced by age in hemodialysis patients, particularly DBP and circulating levels of t-PA, phosphorus, and D-dimer. An adequate evaluation of older patients needs to consider these changes. The interplay of aging and CKD-associated mechanisms seems to justify our observations, but the independent contribution of CKD to the observed age-related changes deserves further clarification.

## 4. Materials and Methods

### 4.1. Patients

All procedures performed in studies involving human participants were in accordance with the 1964 Helsinki Declaration, as revised in 2008. The National Data Protection Commission (Proc. No. 762/2017; Authorization No. 532/2017) and the Ethics Committee of the Faculty of Pharmacy, University of Porto (Report No. 26-04-2016), approved the investigation and data analysis. Informed consent was obtained from all individual participants included in the study. Two hundred eighty-nine (*n* = 289) ESRD adult patients under dialysis therapy for at least 90 days, from 5 dialysis clinics in the Northern region of Portugal, were selected during the period from February to July 2017 and included in a cross-sectional study. Patients were clinically evaluated by nephrologists, and blood was collected for the analytical studies before the midweek dialysis session. Data regarding demographic characteristics, CKD etiology, medical history, dialysis, and pharmacological prescriptions were collected. Patients with known autoimmune disease, active malignancy, and acute or chronic infection were excluded (of 308 patients initially available, 19 were excluded for meeting these excluding criteria, resulting in the 289 selected cases). Patients were classified as older when presenting an age equal to or higher than 65 years, according to PORDATA—The Database of Contemporary Portugal.

Diabetes was defined by the current guidelines (fasting plasma glucose ≥ 126 mg/dL or 2 h plasma glucose ≥ 200 mg/dL during an oral glucose tolerance test or HbA1c ≥ 6.5%) or by the use of antidiabetic agents [46]. Hypertension was defined as a blood pressure equal to or higher than 140/90 mm Hg (average pre-dialysis values from the previous month) or by the use of antihypertensive medication.

### 4.2. Erythropoiesis Stimulating Agents (ESA) and Iron Therapies

Treatment with recombinant human erythropoietin (rhEPO) and with intravenous iron was based on the European Renal Best Practice Guidelines [47].

ESA prescription included three different molecules: epoetin α (Eprex^®^; IU), epoetin β (Neorecormon^®^; IU), and darbepoetin α (Aranesp^®^; μg). The doses of epoetin were converted to standardized equivalent doses of darbepoetin α according to the World Health Organization (WHO) daily-defined dose (DDD): 1000 IU of epoetin is equivalent to 4.5 μg of darbepoetin α (conversion factor: 222:1) [48]. Patients on iron therapy used iron sucrose (Venofer^®^).

### 4.3. Assays

Blood samples, collected immediately before the dialytic procedure, were processed within 2 h. Blood was collected into tubes with and without anticoagulant (K_3_-EDTA) in order to obtain whole blood, plasma, and serum. Aliquots of plasma and serum were immediately stored at −80 °C until assayed.

Erythrocyte, leukocyte, and platelet counts, hemoglobin concentration, and hematocrit values were evaluated by using an automatic blood cell counter (Sysmex K1000; Sysmex, Hamburg, Germany). Reticulocytes were quantified by microscopic counting on blood smears after vital staining with New methylene blue (Reticulocyte stain; Sigma-Aldrich Co. LLC. St. Louis, MO, USA).

Serum iron concentration was determined using a colorimetric method (Iron, Randox Laboratories Ltd., North Ireland, UK), whereas serum ferritin and serum transferrin were measured by immunoturbidimetry (Ferritin and Transferrin, Randox Laboratories Ltd., North Ireland, UK). Transferrin saturation (TS) was calculated by the following formula: TS (%) = 70.9 × serum iron concentration (µg/dL)/serum transferrin concentration (mg/dL). Parathormone concentration was measured in serum by electrochemiluminescence immunoassay (Elecsys PTH, Roche Diagnostics).

Plasma levels of hepcidin and soluble transferrin receptor (sTfR) were evaluated by using standard commercially available enzyme linked immunosorbent assay (ELISA) kits (Human Hepcidin Quantikine ELISA Kit, Human Soluble Transferrin Receptor Quantikine IVD ELISA Kit, R&D Systems, Minneapolis, MI, USA, respectively).

Three conventional inflammatory parameters were quantified in serum: high-sensitivity (hs) C-reactive protein (hsCRP) by immunoturbidimetry (Cardiac C-Reactive Protein (Latex) High Sensitive assay, Roche Diagnostics, Basel, Switzerland), interleukin (IL)-6 and tumor necrosis factor-alpha (TNF-α) by ELISA kits (Human IL-6 Quantikine HS and Human TNF-alpha Quantikine HS, R&D Systems). Plasma pentraxin 3 (PTX3) was evaluated using an ELISA kit (Human Pentraxin 3/TSG-14 Quantikine ELISA Kit, R&D Systems, MI, USA).

The lipid profile, including total cholesterol, low-density lipoprotein cholesterol (LDL-C), high-density lipoprotein cholesterol (HDL-C), and triglycerides, was determined by enzymatic colorimetric tests (Roche Diagnostics, Basel, Switzerland) using routine automated laboratorial procedures (Cobas Integra 400 Plus auto-analyser; Roche Diagnostics, Basel, Switzerland); serum adiponectin and leptin levels and plasma oxidized LDL (oxLDL) were determined with ELISA Kits (Human Total Adiponectin/Acrp30 Quantikine ELISA Kit and Human Leptin Quantikine ELISA, Kit R&D Systems; Oxidized LDL ELISA Kit, Mercodia AB, Uppsala, Sweden).

Several markers related to endothelial (dys)function and/or fibrinolysis were measured: tissue-type plasminogen activator (tPA), plasminogen activator inhibitor type 1 (PAI-1), D-dimer, and asymmetric dimethylarginine (ADMA). These were measured using ELISA kits (Human Tissue Plasminogen Activator ELISA kit, Human PAI1 ELISA kit, D-Dimer ELISA kit, Abcam, Cambridge, UK; ADMA High Sensitive ELISA (EA209/96), DLD Diagnostika GmbH, Hamburg, Germany).

Creatinine, urea, uric acid, and albumin were measured using standardized automated routine assays (Roche Diagnostics, Basel, Switzerland).

Levels of plasma *N*-terminal pro B-type natriuretic peptide (NT-proBNP) and serum tissue inhibitor of metalloproteinase-1 (TIMP-1) (Human proBNP and Human TIMP1 ELISA kits, Abcam, Cambridge, UK) were assessed as cardiac and renal fibrosis markers, respectively.

To evaluate dialysis adequacy, an additional blood sample was obtained after the hemodialysis session to calculate the urea reduction ratio (URR) and the urea clearance index Kt/V (where *K* is the clearance of urea, *t* is the duration of the dialysis session, and *V* is the patient’s urea distribution volume). The URR was calculated with the formula (1 − [Ct ÷ Co]) × 100, in which Ct is post-dialysis and Co is pre-dialysis serum urea concentration. Daugirdas’s formula was used to calculate eKt/V [49].

### 4.4. Statistical Analysis

Kolmogorov–Smirnov analysis was used to test if the data were normally distributed. Those variables showing normal distribution are presented as mean ± standard deviation (SD), and those non-normally distributed are presented as median [interquartile range]. Differences between groups were tested using the chi-squared test and Fisher’s exact test for categorical variables, while Student’s unpaired *t*-test or the Mann–Whitney U test was used for continuous variables.

The strength of the association between the studied variables was estimated by using Spearman’s rank correlation coefficient. To evaluate the independent association of different variables with age, multiple regression analysis was performed after log transformation of the non-normally distributed variables using stepwise selection, with an entry criterion of *p* < 0.05.

Statistical analysis was performed using the IBM Statistical Package for Social Sciences (SPSS, version 27.0, Chicago, IL, USA) for Windows. Statistical significance was accepted at *p* less than 0.05.

## Figures and Tables

**Figure 1 ijms-25-03325-f001:**
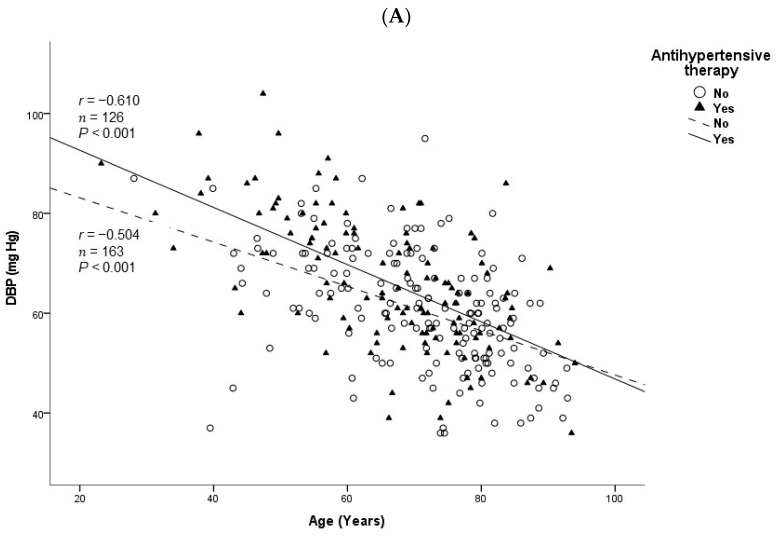
Correlations between the age of the participants and diastolic blood pressure (DBP) according to antihypertensive therapy (**A**) and serum levels of phosphorus according to phosphate binder therapy (**B**) in chronic dialysis patients. *r*, Spearman’s rank correlation coefficient.

**Figure 2 ijms-25-03325-f002:**
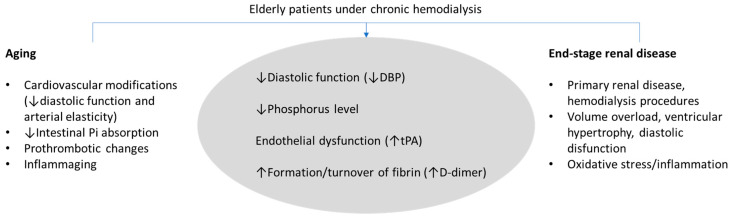
Main causes of age-related biomarker changes in end-stage renal disease (ESRD) patients under chronic dialysis identified in multiple regression analysis. The causes underlying changes in biomarkers in elderly patients are presented, which may have an additive or synergistic effect. Despite pharmacological treatment of hyperphosphatemia in patients, decreased phosphorus levels associated with advanced age. Pharmacological treatment of ESRD patients for complications such as anemia, dyslipidemia, and hypertension explains the lack of change in parameters such as hemoglobin (typically falls), cholesterol, and systolic blood pressure (typically rises) in aged patients. DBP, diastolic blood pressure; tPA, tissue-type plasminogen activator.

**Table 1 ijms-25-03325-t001:** Demographic, clinical, and dialysis-related data in total and in younger and older end-stage renal disease patients under chronic dialysis.

	Total (*n* = 289)	Younger (*n* = 96)	Older (*n* = 193)	*p*
**Age, years**	68.7 ± 13.6	52.8 ± 8.8	76.7 ± 7.0	
**Gender, *n* (%)**				
Male	158 (54.7)	55 (57.3)	103 (53.4)	0.534
Female	131 (45.3)	41 (42.7)	90 (46.6)	
**BMI, Kg/m^2^**	25.8 ± 4.6	25.3 ± 4.8	25.9 ± 4.4	0.282
**Systolic blood pressure (mm Hg)**	137.7 ± 21.0	138.1 ± 23.9	137.5 ± 19.5	0.831
**Diastolic blood pressure (mm Hg)**	63.1 ± 12.8	71.6 ± 12.2	58.8 ± 10.9	**<0.001**
**Etiology of CKD, *n* (%)**				
Diabetic nephropathy	101 (34.9)	26 (27.1)	75 (38.9)	**<0.001 ***
Hypertensive nephrosclerosis	36 (12.5)	11 (11.5)	25 (13.0)	
Polycystic kidney disease	19 (6.6)	9 (9.4)	10 (5.2)	
Chronic glomerulonephritis	23 (8.0)	15 (15.6)	8 (4.1)	
Other or undetermined	110 (38.0)	35 (36.4)	75 (38.8)	
**Dialysis Vintage, Years**	3.74 (1.65–7.40)	3.45 (1.70–7.01)	3.79 (1.61–7.51)	0.967
**Dialysis Therapy, *n* (%)**				
Hemodialysis	41 (14.2)	10 (10.4)	31 (16.1)	0.215 *
On-line hemodiafiltration	248 (85.8)	162 (83.9)	86 (89.6)	
**Vascular Access, *n* (%)**				
Arteriovenous fistula	233 (80.6)	72 (75.0)	161 (83.4)	0.181 *
Arteriovenous graft	14 (4.8)	7 (7.3)	7 (3.6)	
Central venous catheter	42 (14.5)	17 (17.7)	25 (13.0)	
**Intradialytic therapy**				
ESA prescription, *n* (%)	240 (83.0)	78 (81.3)	162 (83.9)	0.618
ESA dose (µg/Kg/week)	0.36 (0.20–0.61)	0.38 (0.22–0.60)	0.35 (0.19–0.61)	0.441
Iron prescription, *n* (%)	186 (64.4)	66 (68.8)	120 (62.2)	0.299
Iron dose (mg/week)	50.0 (25.0–60.0)	50 (25–60)	35 (25–57.5)	0.606
**Most common home medication**				
Statins, *n* (%)	157 (54.3%)	49 (51.0%)	108 (56.0%)	0.454
Antihypertensives, *n* (%)	126 (43.6%)	51 (53.1%)	75 (38.9%)	**0.024**
Antidiabetics, *n* (%)	112 (38.8%)	31 (32.3%)	81 (42.0%)	0.125
**Biochemical and Dialysis Markers**				
Sodium, mEq/L	137 (135–139)	137 (135–139)	137 (136–139)	0.610
Potassium, mEq/L	5.17 ± 0.72	5.23 ± 0.74	5.15 ± 0.72	0.339
Phosphorus, mg/dL	4.18 (3.35–5.00)	4.90 (4.12–5.83)	3.90 (3.30–4.61)	**<0.001**
Calcium, mg/dL	9.10 ± 0.59	8.98 ± 0.65	9.16 ± 0.55	**0.014**
Parathormone, pg/mL	338 (203–502)	359 (193–584)	327 (206–471)	0.600
Uric acid, mg/dL	6.45 ± 1.44	6.99 ± 1.58	6.18 ± 1.28	**<0.001**
Creatinine, mg/dL	8.09 ± 2.34	9.17 ± 2.60	7.55 ± 2.00	**<0.001**
Albumin, g/dL	3.8 (3.6–4.1)	3.9 (3.7–4.1)	3.8 (3.6–4.0)	**0.008**
Urea, mg/dL	116 (97–145)	122 (106–160)	111 (92–141)	**<0.001**
URR, %	79.0 (76.0–83.0)	78.0 (75.3–82.0)	80.0 (76.0–83.0)	0.215
eKt/V	1.63 ± 0.29	1.65 ± 0.32	1.62 ± 0.28	0.435
Ultrafiltration volume, L	2.3 (1.7–2.9)	2.7 (2.0–3.1)	2.2 (1.6–2.8)	**<0.001**

Data presented as mean ± standard deviation or as median (interquartile range), unless otherwise indicated; *p* value for comparisons between younger and older patients; * comparison of frequency distribution of variables. BMI—body mass index; CKD—chronic kidney disease; ESA—erythropoiesis stimulating agents; URR—urea reduction ratio. ESA dose was calculated by converting the doses of epoetin (α and β) to standardized equivalent doses of darbepoetin α according to the World Health Organization (WHO) daily defined dose.

**Table 2 ijms-25-03325-t002:** Hematological and biochemical data in total and in younger and older end-stage renal disease patients under chronic dialysis.

	Total (*n* = 289)	Younger (*n* = 96)	Older (*n* = 193)	*p*
**Hematological Data**				
Erythrocytes (×10^12^/L)	3.72 (3.44–4.01)	3.75 (3.55–3.98)	3.71 (3.43–4.01)	0.397
Hemoglobin (g/dL)	11.4 (10.7–12.2)	11.5 (10.8–12.3)	11.4 (10.6–12.2)	0.524
Hematocrit (%)	35.1 (33.0–37.2)	35.2 (33.4–37.0)	35.1 (32.8–37.4)	0.984
Reticulocytes (×10^9^/L)	40.1 (26.2–57.8)	42.0 (29.7–58.8)	38.9 (25.5–57.4)	0.276
Platelets (×10^9^/L)	197 (161–233)	198 (164–240)	196 (160–232)	0.546
Leukocytes (×10^9^/L)	6.3 (5.3–7.6)	6.3 (5.4–8.0)	6.4 (5.3–7.6)	0.344
**Iron metabolism markers**				
Iron (µg/dL)	55.5 (45.0–74.0)	56.0 (46.0–75.0)	54.0 (45.0–74.0)	0.513
Transferrin (mg/dL)	183.0 (165.0–210.5)	185.0 (167.0–213.0)	180.0 (164.0–209.0)	0.464
Transferrin saturation (%)	21.9 (16.8–28.5)	21.7 (17.4–29.0)	22.0 (16.7–27.6)	0.642
sTfR (nmol/L)	21.6 (16.8–28.0)	22.8 (18.2–28.1)	20.6 (16.1–27.9)	0.105
Ferritin (ng/mL)	342.0 (213.0–484.5)	336.0 (183.0–457.5)	342.0 (218.5–503.0)	0.251
Hepcidin (ng/mL)	78.4 (40.9–131.6)	78.7 (31.8–116.4)	77.5 (43.4–138.0)	0.264
**Endothelial function/fibrinolysis markers**				
tPA (ng/mL)	4.2 (3.0–6.1)	3.4 (2.1–5.5)	4.4 (3.4–6.3)	**<0.001**
PAI-1 (ng/mL)	7.6 (4.7–11.6)	8.1 (4.2–12.9)	7.4 (4.9–10.8)	0.651
D-dimer (ng/mL)	518.0 (364.0–929.5)	397.0 (264.3–705.3)	654.0 (400.0–1030.0)	**<0.001**
ADMA (µM)	1.06 (0.89–1.29)	1.09 (0.91–1.29)	1.06 (0.86–1.28)	0.445
**Inflammatory markers**				
PTX3 (ng/mL)	1.40 (0.99–2.05)	1.28 (0.99–1.85)	1.42 (0.98–2.11)	0.490
hsCRP (mg/dL)	0.37 (0.16–0.81)	0.37 (0.13–0.77)	0.38 (0.17–0.85)	0.409
IL-6 (pg/mL)	4.10 (2.69–7.33)	3.61 (2.42–5.93)	4.63 (3.00–7.62)	**0.003**
TNF-α (pg/mL)	3.38 (2.66–4.59)	3.03 (2.61–4.33)	3.49 (2.70–4.70)	0.066
**Lipid profile and adipokines**				
Total cholesterol (mg/dL)	160.0 (135.0–188.0)	168.0 (136.0–200.3)	156.0 (134.5–185.0)	0.085
Triglycerides (mg/dL)	133.0 (98.0–185.5)	137.5 (98.3–202.8)	129.0 (97.0–173.0)	0.153
HDL-C (mg/dL)	45.3 (38.3–55.7)	44.5 (36.3–56.5)	45.5 (39.0–54.4)	0.888
LDL-C (mg/dL)	79.7 (64.5–107.8)	83.8 (63.6–109.7)	79.4 (64.5–106.3)	0.775
oxLDL (U/L)	44.8 (35.3–57.3)	45.8 (36.2–58.6)	44.6 (34.1–56.8)	0.623
oxLDL/LDL-C ratio	0.054 (0.046–0.067)	0.053 (0.044–0.070)	0.055 (0.047–0.066)	0.820
Adiponectin (µg/mL)	12.20 (7.83–19.23)	12.19 (7.72–19.34)	12.20 (7.89–19.16)	0.877
Leptin (ng/mL)	15.14 (5.11–45.35)	12.33 (3.71–33.43)	17.87 (6.03–49.98)	**0.033**
**Cardiac marker**				
NT-proBNP (ng/mL)	13.55 (8.29–24.73)	10.16 (6.17–17.79)	15.93 (9.36–25.87)	**<0.001**
**Renal fibrosis marker**				
TIMP-1 (ng/mL)	535.0 (468.5–619.5)	516.0 (447.5–575.8)	543.0 (476.0–642.5)	**0.008**

Data presented as median (interquartile range); *p* value for comparisons between younger and older patients. sTfR, soluble transferrin receptor; tPA, tissue-type plasminogen activator; PAI-1, plasminogen activator inhibitor type 1; ADMA, asymmetric dimethylarginine; PTX3, pentraxin 3; hsCRP, high-sensitivity C-reactive protein; IL-6, interleukin-6; TNF-α, tumor necrosis factor-alpha; HDL-C, high-density lipoprotein cholesterol; LDL-C, low-density lipoprotein cholesterol; oxLDL, oxidized LDL; NT-proBNP, N-terminal pro B-type natriuretic peptide; TIMP-1, tissue inhibitor of metalloproteinase-1.

**Table 3 ijms-25-03325-t003:** Statistically significant correlations between age of participants and studied biomarkers for all studied patients and according to gender.

Variable	All Patients (*n* = 289)		Females (*n* = 131)		Males (*n* = 158)	
	*r*	*p*	*r*	*p*	*r*	*p*
DBP	−0.562	<0.0001	−0.587	<0.0001	−0.547	<0.0001
tPA	0.380	<0.0001	0.482	<0.0001	0.301	0.0001
D-dimer	0.367	<0.0001	0.435	<0.0001	0.321	<0.0001
Phosphorus	−0.352	<0.0001	−0.327	0.0001	−0.380	<0.0001
Creatinine	−0.339	<0.0001	−0.297	0.0006	−0.385	<0.0001
Uric acid	−0.319	<0.0001	−0.203	0.0200	−0.381	<0.0001
NT-proBNP	0.279	<0.0001	0.186	0.0339	0.352	<0.0001
UF volume	−0.261	<0.0001	−0.273	0.0016	−0.235	0.0030
Albumin	−0.255	<0.0001	−0.230	0.0083	−0.266	0.0007
TIMP-1	0.227	0.0001	0.314	0.0003	0.153	0.0545
IL-6	0.215	0.0002	0.266	0.0022	0.193	0.0150
Urea	−0.205	0.0005	−0.205	0.0188	−0.193	0.0149
Triglycerides	−0.157	0.0073	0.060	0.4946	−0.305	0.0001

*r*, Spearman’s rank correlation coefficient; DBP, diastolic blood pressure; tPA, tissue-type plasminogen activator; NT-proBNP, *N*-terminal pro B-type natriuretic peptide; UF, ultrafiltration; TIMP-1, tissue inhibitor of metalloproteinase; IL-6, interleukin-6.

**Table 4 ijms-25-03325-t004:** Main variables associated with age in chronic dialysis patients by multiple linear regression analysis.

Dependent Variable	Model	Unstandardized Coefficients		Standardized Coefficients	t	*p*
		B	Std. Error	Beta		
Age (y)	(Constant)	99.215	4.215		23.539	<0.001
	DBP	−0.427	0.049	−0.407	−8.709	<0.001
	Ln tPA	1.243	0.212	0.272	5.865	<0.001
	Ln Phosphorus	−2.606	0.523	−0.230	−4.986	<0.001
	Ln D-dimer	0.002	0.001	0.197	4.200	<0.001

R^2^ for multivariable regression model = 0.510. DBP, diastolic blood pressure; tPA, tissue-type plasminogen activator.

## Data Availability

Data is contained within the article.

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
