# Peer review of "Age-Related Changes in Clinical and Analytical Variables in Chronic Hemodialyzed Patients"

_ijms, 2024, doi:10.3390/ijms25063325_

Round 1

Reviewer 1 Report

Comments and Suggestions for Authors

In the present paper, Belo and co-workers investigated and discussed the age-related changes in different haematological and biochemical parameters in ESRD patients on hemodialysis as a type of RRT. The results and conclusions are based on an extensive data set collected from several hundred patients. The paper is well-written and easy to follow.
I have a few minor comments that authors could consider including and discussing in a revised version of the manuscript.
Section 2.1. second paragraph- consider including cut-off values for diabetes diagnosis, as you also included BP values.
Table 1 - please explain how URR was calculated. In the same table, are p-values missing in the section Etiology of CKD and Vascular Access?
Figure 1 - think of including information on the number of patients on antihypertensive and antidiabetic therapy in Table 1. (how many patients are diagnosed with hypertension and DM).
Think of including (if you have) information on total protein and fibrinogen concentration, as well as calcium concentration and levels of parathormone.
Reference values of measured biochemical and haematological parameters would be helpful for readers.

Reviewer 2 Report

Comments and Suggestions for Authors

I have read and reviewed an interesting paper entitled "Age-related changes in clinical and analytical variables in chronic hemodialyzed patients". Here are my comments:

- abstract section:

    - please change the first phrase "Chronic dialysis populations are getting older throughout the world".

   - These two phrases can be transformed into one : "In multiple linear regression analysis, DBP values, 38 tPA, phosphorus and D-dimer levels were independently associated with the age of patients (standardized betas: -0.407, 0.272, -0.230 and 0.197, respectively; P < 0.001 for all). Our data show changes in several biomarkers with increasing age of dialysis patients, with relevant expressions at cardiovascular and nutritional levels."

- introduction section:

     - "The same applies for D-dimer values in diagnosing deep vein thrombosis in older patients" - please add a reference 

    - please add some information about the end-stage renal disease (ESRD) and chronic dialysis therapy incidence in worldwide population or in Portugal population. Your study is about dialyzed patients, and some information about aging in this specific population should be offered. As you know, not all CKD patients are in need of RRT.  

- material and methods:

    - as you presented the excluded patients, a flowchart containing the screened patients, included and excluded ones should be interesting to see. 

    - how did you choose the cutoff for age for establishing the younger and the older group? Please present this information in this section. 

    - Who clinically assessed the patients? The nephrologist? What parameters did you evaluate? 

    - do you have a national guideline according to which ESRD patients can only use iron sucrose (Venofer®)? 

- results: - 

- discussion:

    - "Contrarily to the general population, we found no association of SBP with age" - not sure about the meaning of this information

    - no information about clinical evaluation results compared to other studies were presented, apart from blood pressure. Maybe you can emphasize this aspect. 

Comments on the Quality of English Language

Moderate editing of English language required. 
